# Wetland eco-engineering: measuring and modeling feedbacks of oxidation processes between plants and clay-rich material

Rémon Saaltink[1], Stefan C. Dekker[1], Jasper Griffioen[1,2], Martin J. Wassen[1]

5   [1] Department of Environmental Sciences, Copernicus Institute of Sustainable Development, Utrecht University, Utrecht 3508 TC, The Netherlands.

[2] TNO Geological Survey of the Netherlands, Princetonlaan 6, 3584 CB Utrecht, The Netherlands

10   **Corresponding author**

Rémon Saaltink

e-mail: r.m.saaltink@uu.nl

tel: +31 30 253 2404

**Abstract**

Interest is growing in using soft sediment as a building material in eco-engineering projects. Wetland construction in the Dutch lake Markermeer is an example: here the option of dredging some of the clay-rich lake-bed sediment and using it to construct wetland will soon go under
20   construction. Natural processes will be utilized during and after construction to accelerate ecosystem development. Knowing that plants can eco-engineer their environment via positive

or negative biogeochemical plant–soil feedbacks, we conducted a six-month greenhouse experiment to identify the key biogeochemical processes in the mud when *Phragmites australis* is used as an eco-engineering species. We applied inverse biogeochemical modeling to link observed changes in pore water composition to biogeochemical processes. Two months after transplantation we observed reduced plant growth and shriveling and yellowing of foliage. The N:P ratios of plant tissue were low and were affected not by hampered uptake of N but by enhanced uptake of P. Subsequent analyses revealed high Fe concentrations in the leaves and roots. Sulfate concentrations rose drastically in our experiment due to pyrite oxidation; as reduction of sulfate will decouple Fe-P in reducing conditions, we argue that plant-induced iron toxicity hampered plant growth, forming a negative feedback loop, while simultaneously there was a positive feedback loop, as iron toxicity promotes P mobilization as a result of reduced conditions through root death, thereby stimulating plant growth and regeneration. Given these two feedback mechanisms, we propose that when building wetlands from these mud deposits Fe-tolerant species are used rather than species that thrive in N-limited conditions. The results presented in this study demonstrate the importance of studying the biogeochemical properties of the building material and the feedback mechanisms between plant and soil prior to finalizing the design of the eco-engineering project.

**Keywords:** Drying; Fe-P; Iron toxicity; P mobilization; PHREEQC; Pyrite

## 1. Introduction

Nowadays natural processes are being used across the world to achieve fast ecosystem development while at the same time providing opportunities for developing hydraulic infrastructure, a concept called Building with Nature (BwN) (Temmerman et al., 2013). Though mostly focused on water safety and coastal protection (e.g. Borsje et al., 2011), BwN can also be applied for the management of fine sediments. A relevant application could be to use soft

sediments as material for building freshwater wetlands. Here, vegetation can be used as an eco-engineer (Jones et al., 1994), to modify the environment (Lambers et al., 2009). When fine sediments are used for the construction of wetlands, however, the use of eco-engineers is anticipated to pose challenges in relation to crest stability, consolidation and soil formation.

In the Netherlands, a soft clay-rich lake-bed sediment in the Markermeer (an artificial lake of

691 km$^2$) is causing serious turbidity problems: primary productivity is impeded and biodiversity in the lake is declining (Vijverberg et al., 2011; Noordhuis et al., 2014). Because the lake is shallow, wind-induced waves frequently induce high bed shear stress, which causes sediment to be resuspended (Vijverberg et al., 2011). To improve the ecological conditions in the lake, it is planned to dredge some of the soft clay-rich sediment and use it to construct approximately

10,000 ha of wetland.

Plants produce root exudates which influence soil formation by enhancing microbiological activity (Holtkamp et al., 2011), biological weathering and nutrient cycling (Taylor et al., 2009; Bradford et al., 2013). An example is the ability of plant roots to mobilize P by ligand exchange

and dissolution of Fe-bound P (Fe-P) by citrate and oxalate excretion (Gerke et al., 2000). Plant

roots may also enhance consolidation processes in substrate by increasing horizontal and

vertical drainage (O'Kelly, 2006).

However, both negative and positive plant–soil feedbacks exist, in which the physical and

chemical properties of the soil affect plant development and vice versa (Ehrenfeld et al., 2005).

Therefore, when looking at soil formation it is important to study the sign and strength of these

plant–soil feedback mechanisms. For example, nutrient conditions co-determine the type of

plant community that develops (e.g. Olde Venterink, 2011), which in turn influences the nutrient

conditions in the soil itself (Onipchenko et al., 2001). As feedback mechanisms differ between

plant species (Ehrenfeld et al., 2005), in order to accelerate ecosystem development it is

essential to decide which eco-engineer is best to introduce in these proto-soils.

De Lucas Pardo (2014) found that the Markermeer mud deposits had a high water content

(20–60% of fresh weight) and were largely anoxic, with oxygen present only in the top 2 mm.

Therefore, when such mud is taken from the lake and spread out in contact with the air,

biogeochemical plant–soil processes related to oxidation and drying of the top soil are expected

to play a significant role. It is intended to use two types of clay-rich deposits as building material

for the wetland. They are the products of a combination of historical and present-day factors.

Prior to 1932, the year in which the dam cutting off the Zuiderzee from the North Sea was

completed, this was a marine environment into which several rivers discharged, including a

branch of the river Rhine (the river IJssel). Hence, a near-shore marine deposit underlies the

present-day soft clay-rich sediment. This soft clay-rich layer is produced by bioturbation and

physical weathering and continuously resuspends as a result of wave action (Van Kessel et al., 2008; De Lucas Pardo et al., 2013). This layer accumulated after 1976, when northward sediment transport was blocked by a second dam that separated Markermeer from IJsselmeer, thus allowing suspended matter to resettle on top of the marine deposit. We can therefore distinguish two layers: an upper disturbed mud layer prone to bioturbation and erosion, and a

relatively undisturbed layer below.

With the aim of identifying the biogeochemical plant–soil feedback processes that occur when oxidation, drying and modification by plants alter the biogeochemical conditions of these two sediment types, thus in turn affecting vegetation development, we set up an experiment to monitor the chemical composition of pore water. Our study has two subsidiary aims: to ascertain

how plants of *Phragmites australis* eco-engineer their environment by expediting biogeochemical processes in the deposits, and to simulate the geochemical differences between disturbed mud and undisturbed clay deposits and relate these to the processes identified from the pore water by using PHREEQC for inverse modeling. In addition, we altered the grain size of the disturbed mud deposit by adding inert sand to see how grain size distribution

impacts pore water chemistry.

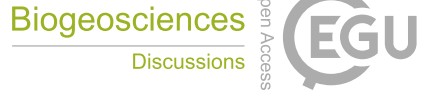

## 2. Material and Methods

### 2.1 Set-up

A greenhouse experiment was conducted for six months at the test facility of Utrecht University. A basin of 4 m² (2 x 2 m) was filled with artificial rainwater and was refreshed every two weeks. At regular intervals, the chemistry of the water was checked to ensure that the water composition remained stable during the experiment. The artificial rainwater was made by adding 15 µmol $NH_4(SO_4)$, 50 µmol $NaNO_3$ and 30 µmol $NaCl$ to osmosis water. These values reflect the

average rainwater composition in the Netherlands for the period 2012–2013 (LMRe, 2014).

The sediments used include the soft clay-rich layer ($Mud_{soft}$) and the underlying, consolidated, Zuiderzee deposit (Clay). In principle, both sediments have the same origin and were collected in the same area. We also included a third sediment type ($Mud_{sand}$), as it is expected that $Mud_{soft}$ will be too soft for use as a building material: a 1:1 mixture was made by mixing mud with

Dorsilit® crystal silica sand (c. 99% $SiO_2$) which had been autoclaved for one hour at 120 °C prior to mixing. The $Mud_{soft}$ and Clay sediments were collected by mechanically dredging in the southern part of the lake and were stored in air-tight containers at 4 °C prior to the start of the experiment.

Plastic pots (diameter 10 cm, depth 18 cm) with a perforated base were filled to within 1 cm

form the top with one of the three sediment types used (t = 0). In each pot, two soil moisture samplers (Rhizon Flex-5cm; Rhizosphere, Wageningen, the Netherlands) were installed horizontally at depths of 1 cm and 11 cm below the sediment surface (these depths are hereafter referred to as $D_1$ and $D_{11}$). The pots were stood in rows in the basin. The water level was

maintained at 9 cm so that the sediment at $D_{11}$ remained saturated while the sediment at $D_1$

could oxidize and dry. Each sediment type had 13 replicates.

Reed seedlings (*Phragmites australis*) had been grown in nutrient-poor peat and when 35–

40 days old (experimental time t = 22 days), a single reed seedling was planted per pot in eight

of the replicates, leaving five replicates unplanted. Any other seedlings that germinated

spontaneously in the pots were removed immediately.


### 2.2 Chemical analysis

Soil moisture at $D_1$ and $D_{11}$ was collected from the moisture samplers on days 0, 3, 10, 22, 36,

64, 92, 134 and 174 from five of the pots per condition. The samples from the five replicates

were pooled and chemically analyzed. Chloride, $NH_4$, $NO_2$, $NO_3$ and $SO_4$ were determined using

ion chromatography (IC); Ca, Fe, K, Mn, Na, P, Si and Sr were determined with Inductively

Coupled Plasma Optical Emission Spectrometry (ICP-OES), pH by an ion-specific electrode,

and alkalinity was measured by a classic titration method.

Sediment samples were collected for each sediment type at t = 0 and were freeze–dried and

stored anoxically prior to geochemical analysis. The major elements were determined using

ICP-OES following an aqua regia destruction. Total S content was measured on an elemental

CS analyzer and the mineralogical composition was determined with X-ray diffraction (XRD). A

sequential extraction method based on Ruttenberg (1992) was applied to characterize solid P

speciation. The method involves five steps (Table 1), the first four of which were carried out

anoxically. Loss on ignition (LOI) was determined by slowly heating to 1000 °C. LOI was also

used as a proxy for organic matter content and total carbonates by calculating the weight loss

between 105–550 °C for organic matter and the weight loss between 550–1000 °C for total

carbonates (Howard, 1965). Cation exchange capacity (CEC) of the sediments was calculated

from the organic matter content and the amounts and types of clay minerals present (Bauer and

Velde, 2014).

Fifty seedlings of *P. australis* randomly chosen from the seedlings grown for the experiment

were used to determine the initial tissue contents of Fe, K, P, and N. Their roots, shoots, and

leaves were separated and air dried. The air–dried material was then ground and analyzed with

total reflection X-ray fluorescence (TXRF) to determine tissue contents of Fe, K, and P. Nitrogen

was determined on an elemental CN analyzer. At the end of the experiment (t = 174), the plants

in the pots were harvested and subjected to the same procedure, to determine the tissue

contents of Fe, K, P, and N.

*2.3 Modeling of biogeochemical processes*

To identify important biogeochemical processes during the incubation experiments, we modeled

with PHREEQC (Parkhurst and Apello, 2013). PHREEQC modeling is frequently used in

geochemical research focusing on issues of water quality: examples include investigating

mineral weathering in a mountain river (Lecomte et al., 2005), deducing geochemical processes

in groundwater (Belkhiri et al., 2010) and investigating the interaction between two aquifers

(Carucci et al., 2012). Here, we applied it to identify biogeochemical plant–soil processes during

the oxidation and natural drying out of the soil.

The model approach is based on mass-balance equations of preselected mineral phases (reactants). The mineral phases can either precipitate (leave the solution) or dissolve (enter the solution) and these are expressed in mole transfers. As we only know the dynamics in concentrations of the pore water, we applied inverse modeling in which all possible

combinations of the mass-balance equations are accepted within a range of measured pore water concentrations ± 4%. We can simulate infiltration or evaporation rates from the pore water. Since in freshwater mud deposits, the dissolution or precipitation of salts (e.g. NaCl) is negligible and can be ignored, the change in pore water Cl concentration was used to calculate the amount of water evaporated or infiltrated.

To enable the model to attribute some of the chemical changes to cation-exchange processes we included an assemblage of exchangers (X): $CaX_2$, $FeX_2$, $KX$, $MgX_2$, $NaX$ and $NH_4X$. The sum of this assemblage was defined as CEC calculated from the sediment composition. CEC is important, since it can buffer some of the biogeochemical processes in sediments by adsorption or desorption of cations.

We used three time frames in our models: 1) oxidation and natural drying out of the soil before the seedlings were transplanted into the pots (t = 0–22 days); 2) initial stage of plant growth (t = 22–64 days); and 3) the stage in which roots started to influence pore water chemistry (t = 64–176 days).

Inverse modeling was applied for all combinations (sediment type, plant/no plant, and depth)

for each time frame. For every combination, several valid simulations were found, due to small


differences in the amount of mole transfers attributed to the mineral phases. Here we present

the plausible simulation with the least amount of mole transfers for each combination.

## 3. Results and Discussion

Below, first the three sediment types will be compared in terms of certain geochemical and

mineralogical elements. Next, the composition of the pore water will be introduced and will be

linked to biogeochemical processes by presenting and discussing the PHREEQC model

simulations. Then, the plant response is presented and discussed in terms of biomass and plant

tissue chemistry. Lastly, the implications for eco-engineering will be discussed.

### 3.1 A brief comparison between sediment types

Table 2 shows the geochemical composition of the disturbed $Mud_{soft}$ and $Mud_{sand}$ and

undisturbed Clay sediments used in this study. The differences between $Mud_{soft}$ and $Mud_{sand}$

are solely attributable to the presence of inert Dorsilit®.

The total sediment concentrations of Al, Fe, Mg, Mn, Na, P, and Zn were significantly higher

in Clay then in $Mud_{soft}$. The quartz content was also higher in Clay, which suggests that there

were more reactive minerals in this type of sediment.

Sequential P extraction revealed that the significant difference in total P consists of a

significantly lower content of Fe-P in $Mud_{soft}$ than in Clay (279 mg/kg versus 772 mg/kg; $p <$

0.01); the other P pools did not differ significantly. The presence of Fe-P in the anoxic Clay

sediment was unexpected, as in anoxic conditions Fe prefers to bind with S to form $FeS_2$. However, after exhaustion of S, precipitation of Fe(II) phosphates may occur (Jilbert and Slomp, 2013). Another possibility is that the reduction of crystalline Fe(III) is not complete in the anoxic

sediment because kinetic processes are slow (Canavan et al., 2007). This is likely the case in Markermeer, given our strict anoxic procedures for storage and analysis of the samples. The exchangeable (or loosely sorbed) P was low in $Mud_{soft}$ and Clay, indicating that only a small part of the total P found in the sediments was readily available for uptake. The other three P-pools were fairly similar and did not differ significantly between the two types of sediment.

The mineralogical analysis (XRD) showed not only that the quartz content was lower in $Mud_{soft}$ than in Clay (37% versus 48%) but that the amounts of calcite and pyrite did not differ between the two types of sediment (9% calcite and 0.6% pyrite). The amount of phyllosilicates (sum of illite, smectite, kaolinite, and chlorite) was higher in $Mud_{soft}$ than in Clay: 43% versus 30%. This must also have caused the CEC to be higher in $Mud_{soft}$, as the organic matter content did not

differ much between the two (7.2% in $Mud_{soft}$ and 6.8% in Clay).

### 3.2 Pore water composition

Figure 1 presents time series for the pore water concentrations of the three macronutrients N, P, and K. The initial decrease in $NH_4$ and increase in $NO_x$ at a depth $D_1$ for the planted conditions

was most likely caused by nitrification as a result of oxidation (Figure 1a–f). At the end of the experiment, almost all dissolved inorganic nitrogen had been removed from the pore water in the pots with plants, whereas in the pots without plants the $NH_4$ concentrations remained

substantial. Furthermore, a high peak of $NO_x$ was observed in Clay sediments at day 10 of the experiment. At a depth $D_{11}$, no large changes were found in general for $NH_4$ and $NO_x$.

A sharp decline in soluble P was visible at $D_1$ for all three sediments, probably because P precipitated with Fe(III) when oxygen penetrated the top layer (Figure 1g–i). However, in Clay this decline was preceded by an increase in P. After several weeks, a thin moss layer started to develop on top of the $Mud_{soft}$ sediment, which probably prevented oxygen from penetrating and thereby increased the P concentrations (Figure 2g). Similar developments were observed for

$Mud_{sand}$ although here the moss layer developed much later. In Clay, no moss grew throughout the experiment.

Concentrations of K were higher than concentrations of N and P and increased in the first few weeks (Figure1j–l). No difference was found between pots at $D_{11}$ with or without plants. However, K was significantly higher at $D_1$ in the planted pots with $Mud_{sand}$ ($p < 0.05$).

Although it may be important to study measured concentrations of nutrients in pore water in order to understand plant functioning, deriving biogeochemical processes from measured data is problematic because it must be taken into account that changes in pore water can be caused by multiple processes such as drying, dilution, dissolution, and precipitation. Figure 2 reveals that the drying of soils at $D_1$ was probably an important factor, because we observed an initial

increase in Cl that indicated that Cl could not dissolve in the three sediments used (e.g. halite dissolution). Drying will have influenced other variables as well, such as sulfate (Figure 2d–f). Comparing the patterns of Cl and $SO_4$ suggests that the change in $SO_4$ concentrations at $D_1$ should be partly attributed to drying out of soils and partly either to dissolution (e.g. pyrite


oxidation) or to precipitation (e.g. gypsum formation). This highlights the need to use

geochemical reaction models like PHREEQC to inversely derive biogeochemical processes

from measured data.

### 3.3 Pore water processes (PHREEQC model simulations)

The main pore water processes modeled by PHREEQC are presented in Table 3. For clarity,

only major reactants are included in this Table. Supplementary Tables A1 and A2 present mole

transfers for all reactants used, as well as the number of valid simulations per combination

found.

### 3.3.1 Phase 1: Oxidation and drying (t = 0–22 days)

As discussed in section 3.2, initial drying of soils occurred at $D_1$ immediately after exposure to

air. In the model, this is illustrated by high evaporation rates expressed as $H_2O$ loss (2300–3400

mmol $l^{-1}$ day$^{-1}$; Table 3). The model accounts for this loss by adjusting the solution fractions

before calculating other mole transfers.

Exposure to air also leads to oxidation, more so at $D_1$ than at $D_{11}$ (Table 3). The increase in

measured sulfate is partly explained as pyrite oxidation (109–270 µmol $l^{-1}$ day$^{-1}$ for $D_1$ and 20.1–

36.2 µmol $l^{-1}$ day$^{-1}$ for $D_{11}$, respectively). Oxidation of pyrite also produces iron oxyhydroxides

and protons which in turn promotes dissolution of calcite. The overall reactions are

$$FeS_2 + 3.75O_2 + 3.5H_2O \rightarrow Fe(OH)_3 + 2SO_4^{2-} + 4H^+ \qquad (1)$$



followed by calcite dissolution

$$CaCO_3 + H^+ \rightarrow Ca^{2+} + HCO_3^- \hspace{4cm} (2)$$

The mole transfers for pyrite and calcite presented in Table 3 indicate that not enough calcite is dissolved to buffer all $H^+$ produced by dissolution of pyrite. Indeed, a drop in pH was observed

at the beginning of the experiment (not shown). However, the mineralogical composition presented in Table 2 shows that the amount of calcite (9%; 900 mmol) far exceeds that of pyrite (0.6%; 50 mmol). These numbers suggest that even if all pyrite were to be oxidized, enough calcite is present to buffer all $H^+$ produced (200 mmol). Note that for $Mud_{sand}$ these values are lower due to mixing with Dorsilit®.

Some aeration occurred at $D_{11}$. The $O_2$ fluxes ranged between 61 and 119 $\mu$mol $l^{-1}$ day$^{-1}$, which resulted in small amounts of pyrite being oxidized (20–36 $\mu$mol $l^{-1}$ day$^{-1}$). However, sulfate concentrations did not rise, as a result of subsequent precipitation with Ca to form gypsum (53–73 $\mu$mol $l^{-1}$ day$^{-1}$).

Furthermore, the cation-exchange-capacity (CEC) of the sediments buffered some processes

in pore water chemistry by net adsorption of cations at $D_1$ and net desorption at $D_{11}$.

The processes described above occurred in all three sediments, although some differences were noted. Oxidation was higher in $Mud_{soft}$ than in $Mud_{sand}$ and Clay, probably because higher evaporation rates in $Mud_{soft}$ enhanced oxidation and affected other reactants related to oxidation.


### 3.3.2 Phase 2: Initial stage of plant growth (t = 22–64 days)

While the pore water compositions did not show clear differences between unplanted and planted conditions, the inverse modeling provided clear evidence for differences at $D_1$. However, chemical differences between unplanted and planted conditions for Mud$_{sand}$ might simply be attributed to concentration/dilution due to $H_2O$ loss/gain (-996 to 380 mmol l$^{-1}$ day$^{-1}$).

Overall, more pyrite was oxidized in the planted conditions, though the rates are much lower than in the first phase (0–64.3 µmol l$^{-1}$ day$^{-1}$). This observation provides evidence that plants may enhance pyrite oxidation by radial oxygen loss (i.e. root aeration). Ferric oxide production on pyrite surfaces probably impeded further oxidation of pyrite, which is a common phenomenon in carbonate-buffered conditions (Nicholson et al., 1990). Indeed, the total pyrite that had oxidized after 64 days (6.3 mmol for Mud$_{soft}$, 2.5 mmol for Mud$_{sand}$ and 6.2 mmol for Clay, calculated from the rates presented in Table 3) corresponds to a small fraction of total pyrite present (50 mmol).

Saturation with gypsum led to precipitation of $SO_4$ and Ca at $D_1$. Table 3 shows that with the exception of Mud$_{sand}$, mole transfers were lower for planted conditions; the probable reason is that citric acid production by root tips retarded gypsum precipitation (Prisciandaro et al., 2005). This process was not relevant at $D_{11}$, as here aeration (and subsequent sulfate production) by plant roots was minor (in the case of Clay) or absent (in the case of Mud$_{soft}$ and Mud$_{sand}$).

The thin moss layer that started to develop after several weeks in the unplanted condition on top of the Mud$_{soft}$ sediment slowed down the aeration rate to 2.62 µmol l$^{-1}$ day$^{-1}$ and might be

the reason for the moderate increase in P, which probably resulted from $Fe(OH)_3$ dissolution (0.95 µmol $l^{-1}$ $day^{-1}$) (Figure 1g, Table 3).

### 3.3.3 Phase 3: Root influence (t = 64–176 days)

Phase 3 took place in the autumn, when temperatures were lower and therefore the soils did not dry out; hence there was a net gain in $H_2O$. The gain was less in planted conditions, due to uptake of water by roots.

The fully grown plants continued to influence pore water chemistry at $D_1$, but in the unplanted conditions the chemical changes were minor (Table 3). Radial oxygen loss continued the

oxidation processes described in the previous sections. It should be noted that *P. australis* is known to have higher radial oxygen loss than other wetland species (Brix et al., 1996; Dickopp et al., 2011; Smith and Luna, 2013), so the aeration effect found in this study cannot be assumed to hold for other species.

In contrast to the previous phase, in phase 3 the influence of roots was clearly visible at $D_{11}$

for all three sediments. All planted sediments showed increased aeration and subsequent oxidation of pyrite due to radial oxygen loss, with a notable difference between $Mud_{soft}$ (lower) and $Mud_{sand}$ (higher). This is somewhat surprising, as the belowground biomass was significantly higher in $Mud_{soft}$ (section 3.4). It indicates that increasing the average grain size by adding sand enhanced aeration, even when root biomass production was low.


### 3.4 Plant response

Above- and belowground biomass were significantly higher in $Mud_{soft}$ and Clay than in $Mud_{sand}$ (Figure 3; $p < 0.02$). The difference between the two Mud sediments cannot be explained by nutrient concentrations in pore water or light conditions in the greenhouse, as these were the

same for the two sediments. As biomass production in $Mud_{sand}$ was not limited by chemical or biological properties relative to $Mud_{soft}$, it seems likely that the reason for the lower biomass production in $Mud_{sand}$ is a difference in physical properties. Voorhees et al. (1975) and Bengough and Mullins (1990) showed that so-called mechanical impedance (i.e. the resistance to penetration by the root tip) was higher in loamy sand than in clay, which was attributed to the

higher bulk density of the loamy sand. Therefore, increasing the bulk density of $Mud_{soft}$ by mixing with sand increased the mechanical impedance and this might explain the lower biomass production we observed in $Mud_{sand}$.

   *P. australis* invested more in its root system than in its shoots and leaves for all sediments (Figure 3). More investment in roots implies a limitation of N, P, and/or S (Ericsson, 1995;

Shipley and Meziane, 2002). Figures 1a–i and 2d–f show that the N and P concentrations were indeed low in the planted conditions but that $SO_4$ was high, which rules out S limitation. During the experiment we had observed reduced plant growth and shriveling and yellowing of foliage 2 months after transplantation, which might have been caused by nutrient limitation.

   Figure 4 shows the N, P, and K contents as well as the N:P ratio for the roots of *P. australis*

at the beginning and end of the experiment for the three sediment types. The N, P, and K contents in the roots increased in time, while the N:P ratio clearly decreased. The reduction in



N:P ratio from 11 to 2–3 suggests N was the limiting nutrient as an N:P ratio of < 14 in plant tissue is indicative of N limitation (Koerselman and Meuleman, 1996). However, root N and P concentrations of *P. australis* should typically range between 0.64–1.04% for N and 0.06–0.13%

for P (Wang et al., 2015). Figure 4 shows that the root N and P concentrations were above these values, and that P was particularly high: by a factor of 5 to 10 (N: 1.14–1.63% and P: 0.52–0.62%). Hence the concentrations of these nutrients in the roots do not indicate that nutrient limitation is a likely cause of the reduced plant growth and shriveling and yellowing of foliage.

We hypothesize that co-precipitation of P with Fe on roots enhanced the concentrations of P

in the plant roots (Snowden and Wheeler, 1995; Jørgenson et al., 2012). Snowden and Wheeler (1995) showed that this so-called iron plaque formation enhances uptake of Fe and P. This may cause iron toxicity and is probably responsible for the elevated P concentrations in tissue, and for the stunted growth and leaf decay we observed in the experiment. Note that the plant roots of *P. australis* initiate this process by oxidizing their environment and thereby enabling ferrous

iron to oxidize into P-bearing ferric iron, which precipitates on roots.

The Fe concentration in the leaves and in the roots supports the "Fe-P co-precipitation hypothesis": we measured an approximately 20-fold increase by comparison with the initial concentration in the seedlings (Figure 5). Furthermore, ferric oxide, a product of pyrite oxidation, precipitates on root surfaces (Jørgenson et al., 2012), and hence pyrite oxidation in sediments

is directly linked to iron toxicity in plants.

Further evidence to support our hypothesis is provided by the results of the sequential phosphorus extraction conducted on the sediments: it revealed that the dominant P pool in the

sediments is the Fe-P fraction (Table 2). P co-precipitates with Fe on roots if it is bound to ferric oxides.


### 3.5. Implications for eco-engineering

Our results strongly point in the direction of iron toxicity as a major bottleneck prohibiting healthy development of *P. australis*. Since the candidate material for the construction of the Markermeer wetland has high contents of Fe and Fe-P, we recommend using Fe-tolerant plant species as

test species in the new wetland, rather than species optimized for growing in N-limited conditions.

Concomitantly with iron toxicity, a high Fe-P content in soil will trigger P mobilization if that soil is rewetted after having dried out and contains high amounts of $SO_4$ (Smolders and Roelofs, 1993; Lucassen et al., 2005). In some cases, this can result in elevated levels of sulfide, thereby

promoting S toxicity in plants (Lamers et al., 1998; Van der Welle et al., 2007).

Figure 6 summarizes the important feedbacks and processes we expect play an important role in the clay-rich sediments. Following the feedback loops between plant and soil, we see a negative feedback loop that arises because plant roots induce aeration, which promotes iron toxicity that decreases plant growth and results in plant death. Also, we see a positive feedback

loop, as iron toxicity induces reduction processes as a result of root death, which leads to P mobilization and hence enhances plant growth and regeneration. Negative feedback loops diminish or buffer changes, whereas a positive feedback loop amplifies changes. So, a negative feedback loop normally stabilizes the system, in our case via the toxic effect of iron oxides on

plants, but plant growth may increase due to the positive feedback loop via P mobilization. The

relative strengths of these two feedback loops and the sensitivity of species to Fe toxicity

determine the ultimate effect on vegetation development in wetlands built from these sediments.

As drying–rewetting cycles are likely to occur in these future wetlands and since the Fe-P

concentrations in the building material are high, these feedbacks might be an important factor

influencing soil formation and ecosystem development. We therefore recommend studying the

ultimate effects of the use of this material on ecosystem development by testing with various

plant species and drying–rewetting cycles.

## 4. Conclusions

The results of this study show that plants expedite biogeochemical processes by oxidizing and

modifying their environment, which in turn affects the growth conditions of the plants. In the mud

deposits from Markermeer, the key processes influencing pore water chemistry are pyrite

oxidation and associated calcite dissolution. The former is especially likely to be important as it

is linked to iron toxicity and P mobilization and thus has the potential to initiate two feedback

mechanisms between plant and soil. We found strong indications for a negative feedback loop,

where plant-induced iron toxicity is hampering plant growth and a positive feedback loop, where

iron toxicity promotes P mobilization and thereby enhances plant growth. The strength of these

feedbacks and the balance between them will play an important role in regulating eco-

engineering conditions for plants.

© ①

We found conclusive evidence that the low N:P ratio found in plant tissue was not caused by N limitation, as the ratio suggests, but  probably results from enhanced P uptake, as a result of co-precipitation with Fe on roots.

The magnitude of the feedback mechanisms is expected to differ between the building materials used. The soft clay-rich layer has less Fe-P than the underlying clay layer and
therefore P mobilization is expected to be less in mud. However, when the mud is mixed with sand, the enhanced aeration brought about by the change in grain-size composition results in higher oxidation rates, increasing the impact on P mobilization and iron toxicity.

To study the effects of iron toxicity and P mobilization in greater detail we recommend further testing with different plant species and drying–rewetting cycles. This is important because we
expect these mechanisms to influence soil formation and ecosystem development in the future wetland.

### Acknowledgements

This study was supported with funding from Netherlands Organization for Scientific Research
(NWO), project no. 850.13.032 and the companies Boskalis and Van Oord. We would also like to thank Botanical Garden Utrecht for their help, support and advice during the greenhouse experiment. Joy Burrough advised on the English.



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



**Table 1**. List of steps used in the extraction procedure of phosphorus (based on Ruttenberg, 1992).

| Step | | Extractant | Separated P fraction |
|---|---|---|---|
| I | | 1M MgCl$_2$, 30 min | Exchangeable or loosely sorbed P |
| II | A | Citrate-dithionite-bicarbonate (CDB), 8 h | Easily reducible or reactive ferric Fe- P |
| | B | 1M MgCl$_2$, 30 min | |
| III | A | Na acetate buffer (pH 4), 6 h | Amorphous apatite and carbonate P |
| | B | 1M MgCl$_2$, 30 min | |
| IV | | 1M HCl, 24 h | Crystalline apatite and other inorganic P |
| V | | Ash at 550 $^o$C, 2h; 1M HCl, 24 h | Organic P |




**Table 2.** Geochemical and mineralogical composition of the sediment types used in this study. Significant differences between Mud$_{soft}$ and Clay are indicated by * (p < 0.05).

| | Unit | n per type | Mud$_{soft}$ Mean | SD | Mud$_{sand}$ Mean | SD | Clay Mean | SD |
|---|---|---|---|---|---|---|---|---|
| *Aqua regia / CS* | | | | | | | | |
| **Al\*** | mg/kg | 15 | 16593 | 3130 | 6394 | 2439 | 21989 | 4512 |
| **Ca** | mg/kg | 15 | 45635 | 6020 | 18877 | 3572 | 48031 | 3032 |
| **Fe\*** | mg/kg | 15 | 20745 | 2987 | 7804 | 2281 | 27766 | 3764 |
| **K** | mg/kg | 15 | 4102 | 641 | 1723 | 742 | 5371 | 1262 |
| **Mg\*** | mg/kg | 15 | 6636 | 906 | 2531 | 558 | 8041 | 1017 |
| **Mn\*** | mg/kg | 15 | 577 | 160 | 238 | 62 | 710 | 166 |
| **Na\*** | mg/kg | 15 | 526 | 158 | 219 | 64 | 992 | 379 |
| **P\*** | mg/kg | 15 | 649 | 169 | 259 | 56 | 1186 | 217 |
| **S** | mg/kg | 15 | 5586 | 698 | 3001 | 846 | 5727 | 710 |
| **Sr** | mg/kg | 15 | 135 | 26 | 62 | 14 | 148 | 21 |
| **Ti** | mg/kg | 15 | 312 | 77 | 125 | 44 | 312 | 74 |
| **Zn\*** | mg/kg | 15 | 110 | 29 | 43 | 18 | 159 | 58 |
| | | | | | | | | |
| *Seq. P extraction* | | | | | | | | |
| **Exchangeable P** | mg/kg | 15 | 11.9 | 3.50 | 5.9 | 1.79 | 14.3 | 6.81 |
| **Fe- bound P\*** | mg/kg | 15 | 279 | 61.7 | 94.5 | 29.0 | 772 | 263 |
| **Ca-bound P** | mg/kg | 15 | 121 | 30.9 | 36.8 | 13.1 | 146 | 43.3 |
| **Detrital P** | mg/kg | 15 | 169 | 14.1 | 51.5 | 10.9 | 147 | 16.5 |
| **Organic P** | mg/kg | 15 | 117 | 25.1 | 47.7 | 8.38 | 99.6 | 20.0 |
| | | | | | | | | |
| *XRD* | | | | | | | | |
| **Quartz** | % | 1 | 37 | | n.a. | | 48 | |
| **Calcite** | % | 1 | 9 | | n.a. | | 9 | |
| **Pyrite** | % | 1 | 0.6 | | n.a. | | 0.6 | |
| **Illite** | % | 1 | 21 | | n.a. | | 15 | |
| **Smectite** | % | 1 | 14 | | n.a. | | 11 | |
| **Kaolinite** | % | 1 | 5 | | n.a. | | 3 | |
| **Chlorite** | % | 1 | 3 | | n.a. | | 2 | |
| | | | | | | | | |
| *Other* | | | | | | | | |
| **Organic matter** | % | 5 | 7.2 | 0.6 | 2.8 | 0.4 | 6.7 | 0.6 |
| **CEC (calculated)** | meq/100g | | 37.2 | | 12.4 | | 30.0 | |



**Table 3.** Main pore water processes expressed in mole transfers (µmol l$^{-1}$ day$^{-1}$) as modeled by PHREEQC with pore water data retrieved at 1 cm and 11 cm below sediment surface ($D_1$ and $D_{11}$ respectively). Positive values indicate dissolution, negative values indicate precipitation. Cation exchange capacity (CEC) is the sum of Ca, Fe, K, Mg, Na, and NH$_4$.

| Phase | Condition | | Calcite $D_1$ | Calcite $D_{11}$ | Gypsum $D_1$ | Gypsum $D_{11}$ | Fe(OH)$_3$ $D_1$ | Fe(OH)$_3$ $D_{11}$ | Pyrite $D_1$ | Pyrite $D_{11}$ | ΣCEC $D_1$ | ΣCEC $D_{11}$ | H$_2$O (x 10$^3$) $D_1$ | H$_2$O (x 10$^3$) $D_{11}$ | O$_2$ $D_1$ | O$_2$ $D_{11}$ |
|---|---|---|---|---|---|---|---|---|---|---|---|---|---|---|---|---|
| **1. Oxidation t=0-22 days** | **Mud$_{soft}$** | **No plant** | 267 | 111 | 0.00 | -72.5 | -277 | 0.00 | 270 | 36.2 | -31.3 | 20.2 | -3364 | 0.00 | 1009 | 119 |
| | **Mud$_{sand}$** | **No plant** | 0.00 | 59.6 | 0.00 | -40.7 | -116 | 0.00 | 109 | 21.7 | -4.99 | 7.92 | -2591 | 0.00 | 432 | 69.5 |
| | **Clay** | **No plant** | 120 | 55.2 | 0.00 | -53.4 | -160 | 0.00 | 159 | 20.1 | -91.4 | 14.0 | -2364 | 0.00 | 659 | 61.9 |
| **2. Initial root development t=22-64 days** | **Mud$_{soft}$** | **No plant** | 27.1 | 0.00 | -236 | 0.00 | 0.95 | -0.24 | 0.00 | 0.00 | -23.1 | 1.43 | 0.00 | 0.00 | 2.62 | 0.00 |
| | | **Plant** | 48.8 | 19.8 | -208 | -3.81 | -10.0 | -6.19 | 9.76 | 0.00 | -7.63 | 1.43 | 0.00 | 0.00 | 45.5 | 0.00 |
| | **Mud$_{sand}$** | **No plant** | 39.3 | 71.7 | 0.00 | 0.00 | 0.00 | -41.2 | 0.21 | 0.00 | 1.90 | 1.46 | 380 | 0.00 | 0.00 | 0.00 |
| | | **Plant** | 7.10 | 83.8 | -83.4 | 0.00 | 0.00 | -51.2 | 3.58 | 0.00 | 5.40 | 3.40 | -996 | 0.00 | 0.00 | 0.00 |
| | **Clay** | **No plant** | 0.00 | 27.1 | -32.1 | 0.00 | -21.4 | -25.0 | 21.2 | 0.00 | 0.01 | -0.23 | -286 | 0.00 | 41.9 | 0.00 |
| | | **Plant** | 36.9 | 16.2 | 0.00 | 0.00 | -14.3 | 0.00 | 64.3 | 11.9 | 28.4 | 4.53 | -6.67 | 0.00 | 186 | 40.5 |
| **3. Root influence t=64-176 days** | **Mud$_{soft}$** | **No plant** | 0.00 | -3.21 | -19.2 | 0.00 | -1.34 | -0.80 | 0.00 | 0.00 | -1.07 | -1.43 | 56.3 | 0.00 | 0.00 | 0.00 |
| | | **Plant** | 25.8 | 0.00 | 0.00 | 0.00 | -4.20 | 0.00 | 23.8 | 4.11 | 7.88 | -4.65 | 49.1 | 0.00 | 83.6 | 13.6 |
| | **Mud$_{sand}$** | **No plant** | 8.13 | 0.00 | -7.59 | 0.00 | -10.6 | -1.34 | 0.00 | 0.00 | -1.78 | 1.42 | 74.1 | 0.00 | 0.00 | 0.00 |
| | | **Plant** | 0.00 | 0.00 | -14.8 | 0.00 | -13.3 | -23.2 | 13.8 | 7.95 | 0.12 | -10.6 | -357 | -652 | 44.7 | 32.6 |
| | **Clay** | **No plant** | 0.00 | 11.5 | 0.00 | 0.00 | 0.00 | -13.8 | 33.3 | 0.00 | 23.9 | 0.36 | 134 | 0.00 | 113 | 0.00 |
| | | **Plant** | 115 | 18.7 | 0.00 | 0.00 | -58.5 | -8.48 | 58.3 | 8.57 | 45.4 | -5.73 | 0.00 | -98.2 | 215 | 28.4 |

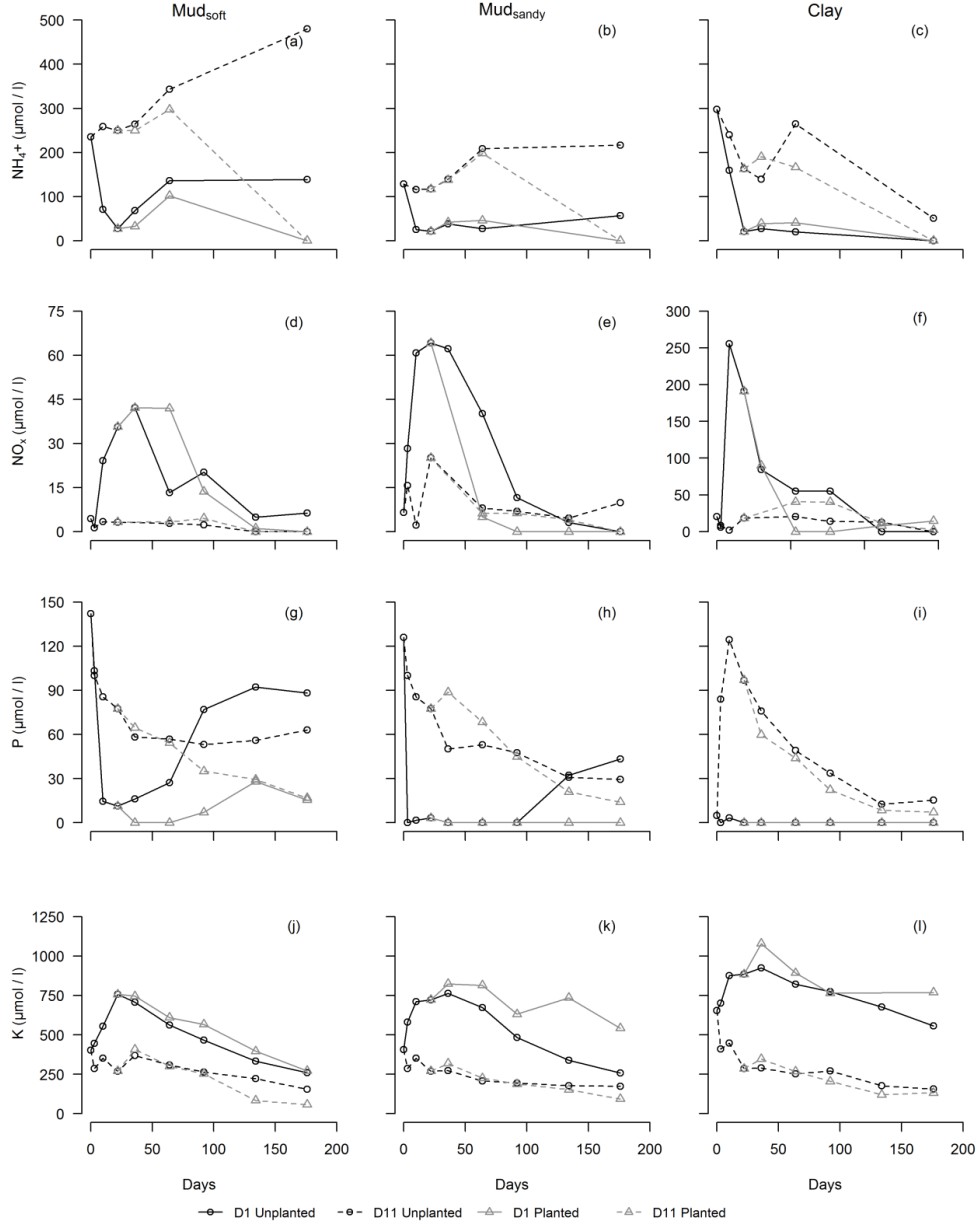

**Figure 1.** Time series of $NH_4$ (a–c), $NO_x$ (d–f), P (g–i) and K (j–l) concentrations. Each column represents one sediment type: $Mud_{soft}$ (a, d, g, j), $Mud_{sand}$ (b, e, h, k), and Clay (c, f, i, l). The variable and the scale of the x-axis are the same for each row, except for the scale in f.





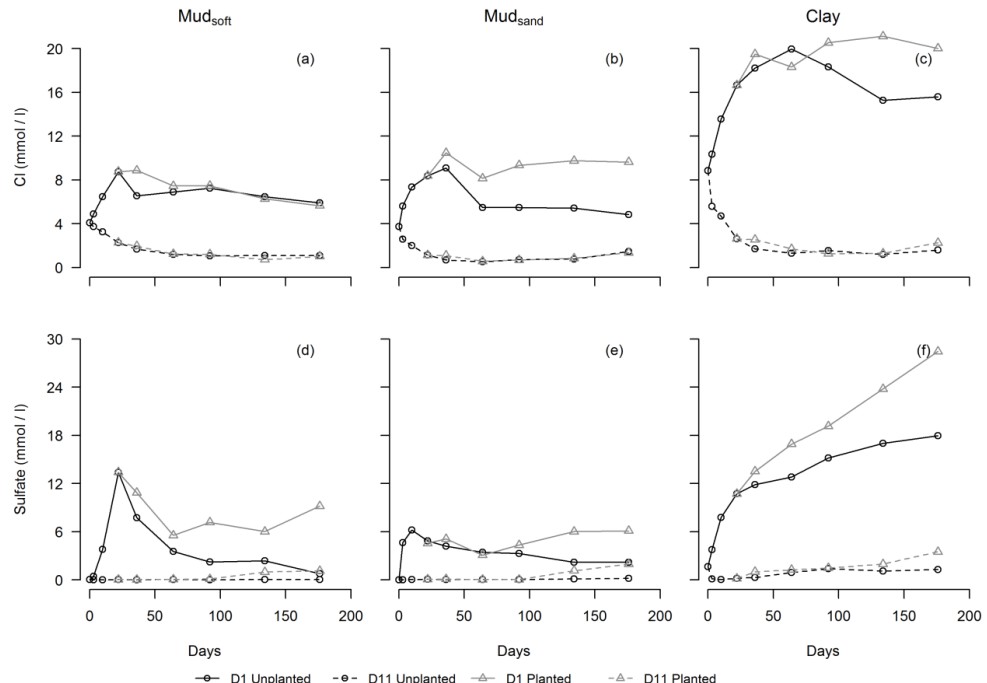

**Figure 2.** Time series of Cl (a–c), and sulfate (d–f) concentrations. Each column

represents one sediment type: Mud$_{soft}$ (a, d), Mud$_{sand}$ (b, e), and Clay (c, f). The

variable and the scale of the x-axis are the same for each row.





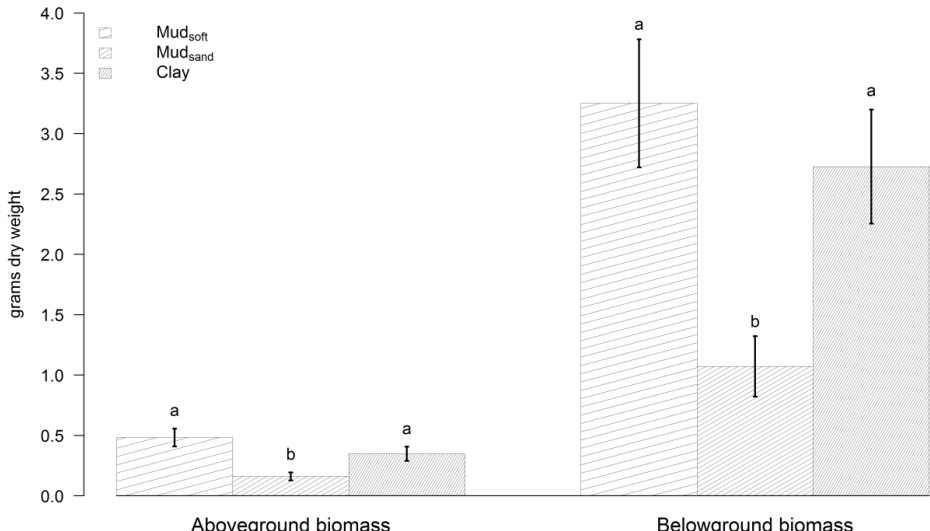

**Figure 3.** Above- and belowground biomass in grams dry weight, with error bars (n =

5). Significant differences between sediment types are indicated by different letters,

and non-significant differences are indicated by a similar letter.



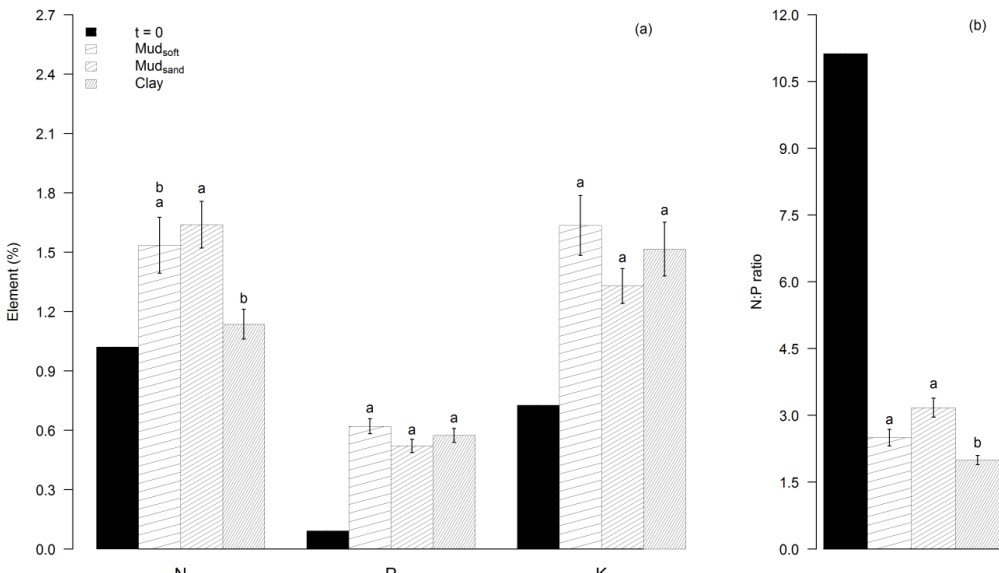

**Figure 4.** N, P, and K concentration in root tissue (t = 176) in % of dry weight (a) as well as the N:P ratio (b) with error bars when n = 5. Significant differences between sediment types are indicated by different letters, and non-significant differences are indicated by a similar letter.





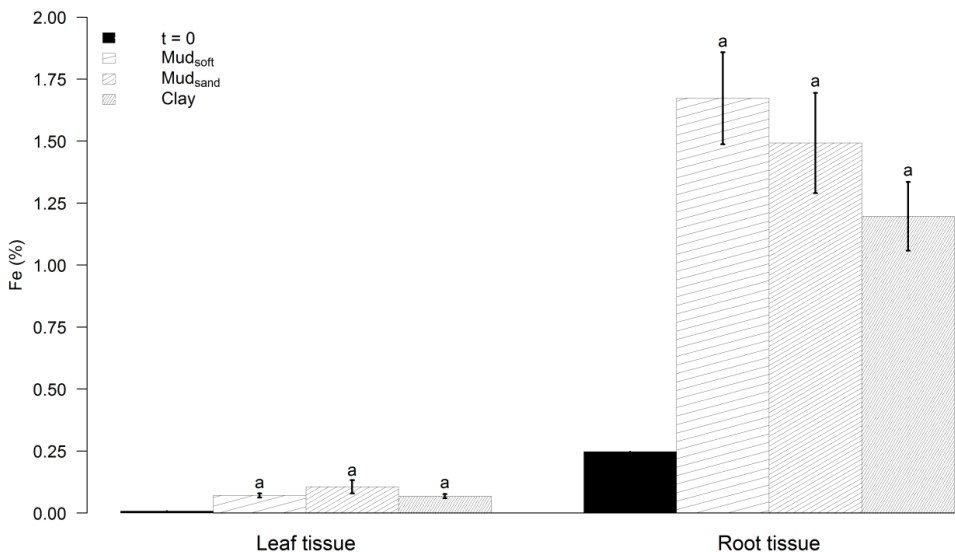

**Figure 5.** Fe concentration (% of dry weight) in leaf and root tissue with error bars
when n = 5. Significant differences between sediment types are indicated by different
letters, and non-significant differences are indicated by a similar letter.



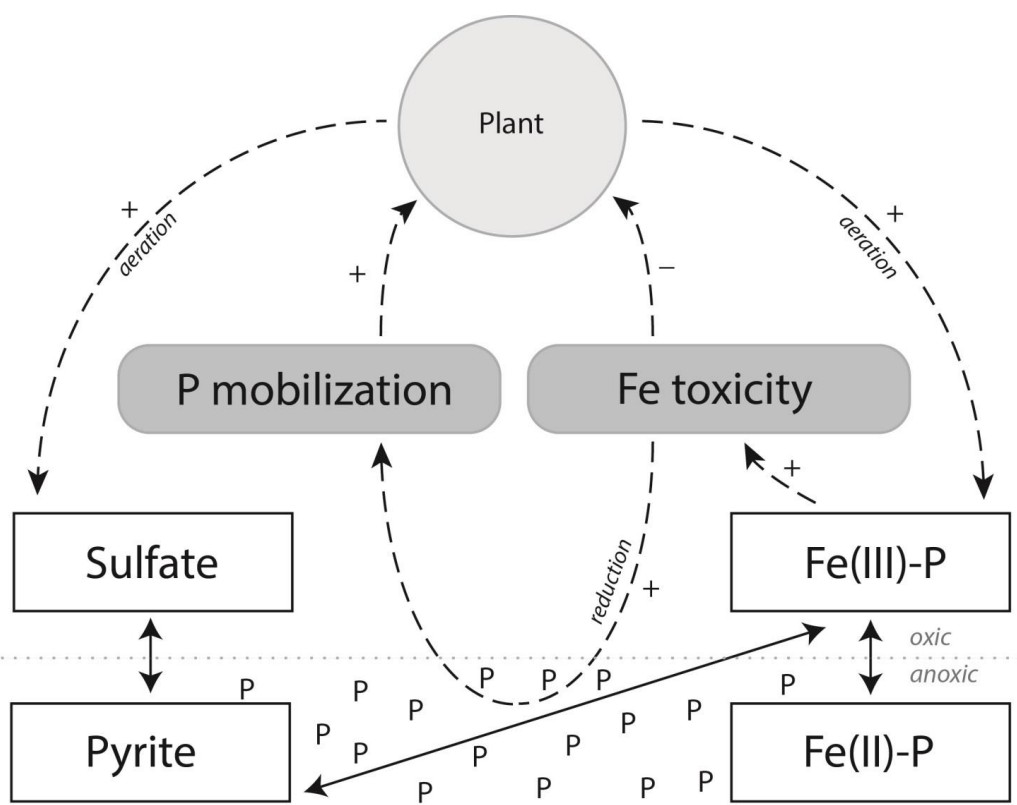

**Figure 6.** Most important biogeochemical processes and feedbacks identified in this study. + indicates positive feedback, - indicates negative feedback.



**Appendix**

**Table A1.** Pore water processes expressed in mole transfers (µmol l$^{-1}$ day$^{-1}$) as modeled by PHREEQC with pore water data

retrieved at 1 cm below sediment surface. Positive values indicate dissolution, negative values indicate precipitation.

| Reactant | Composition | Phase 1. Oxidation (t=0-22) | | | Phase 2. Initial root development (t=22-64) | | | | | | Phase 3. Root influence (t=64-176) | | | | | |
|---|---|---|---|---|---|---|---|---|---|---|---|---|---|---|---|---|
| | | No plant Mud$_{soft}$ | No plant Mud$_{sand}$ | No plant Clay | No plant Mud$_{soft}$ | Plant | No plant Mud$_{sand}$ | Plant | No plant Clay | Plant | No plant Mud$_{soft}$ | Plant | No plant Mud$_{sand}$ | Plant | No plant Clay | Plant |
| Calcite | CaCO$_3$ | 267 | 0.00 | 120 | 27.1 | 48.8 | 39.3 | 7.1 | 0.00 | 36.9 | 0.00 | 25.8 | 8.13 | 0.00 | 0.00 | 115 |
| Gypsum | CaSO$_4$:2H$_2$O | 0.00 | 0.00 | 0.00 | -236 | -208 | 0.00 | -83.4 | -32.1 | 0.00 | -19.2 | 0.00 | -7.59 | -14.8 | 0.00 | 0.00 |
| Hydroxyapatite | Ca$_5$(PO$_4$)$_3$(OH) | -5.00 | -3.64 | 0.00 | 0.24 | 0.00 | -0.02 | -0.04 | 0.00 | 0.00 | 0.18 | 0.00 | 0.09 | 0.00 | 0.00 | 0.00 |
| Chalcedony | SiO$_2$ | -19.1 | -15.5 | -18.2 | 0.95 | 0.71 | 1.91 | -3.37 | -1.67 | -2.14 | 0.71 | 0.00 | 0.54 | 1.43 | 0.00 | -0.36 |
| Fe(OH)$_3$ (a) | Fe(OH)$_3$ | -277 | -116 | -160 | 0.95 | -10.0 | 0.00 | 0.00 | -21.4 | -14.3 | -1.34 | -4.20 | -10.6 | -13.3 | 0.00 | -58.5 |
| Pyrite | FeS$_2$ | 270 | 109 | 159 | 0.00 | 9.76 | 0.21 | 3.58 | 21.2 | 64.3 | 0.00 | 23.8 | 0.00 | 13.8 | 33.3 | 58.3 |
| Rhodochrosite | MnCO$_3$ | -11.8 | -11.4 | -2.27 | 2.86 | 1.19 | 1.23 | 0.34 | -0.24 | -0.24 | -0.63 | -0.89 | 0.09 | 0.18 | 0.00 | 0.00 |
| CEC | CaX$_2$ | 0.00 | 20.9 | 55.5 | 63.1 | 41.9 | -9.11 | 0.00 | 0.00 | 0.00 | 2.50 | 0.00 | -9.73 | 0.00 | -9.64 | -85.4 |
| | FeX$_2$ | 0.00 | 0.00 | 0.00 | 0.00 | 0.00 | -0.19 | -4.11 | 0.00 | -50.2 | 1.61 | -19.8 | 11.7 | 0.00 | -33.3 | 0.00 |
| | KX | -8.64 | -5.00 | -17.7 | -4.76 | 0.00 | 3.78 | -8.30 | -6.19 | 0.00 | -2.14 | -2.14 | -2.77 | -7.68 | 0.00 | 0.00 |
| | MgX$_2$ | 31.4 | -16.8 | 36.8 | -39.8 | -30.5 | 7.42 | -1.35 | 0.00 | 21.7 | -3.04 | 12.0 | 0.00 | 0.00 | 19.1 | 39.8 |
| | NaX | -20.9 | 0.00 | -166 | -46.4 | -25.7 | 0.00 | 25.1 | 25.2 | 77.6 | 0.00 | 19.7 | 0.00 | 12.0 | 49.4 | 92.9 |
| | NH$_4$X | -33.2 | -4.09 | 0.00 | 4.76 | 6.67 | 0.00 | -5.94 | -19.0 | -20.7 | 0.00 | -1.88 | -0.98 | -4.20 | -1.70 | -1.88 |
| H$_2$O (g) | H$_2$O x 10$^3$ | -3364 | -2591 | -2364 | 0.00 | 0.00 | 380 | -996 | -286 | -6.67 | 56.3 | 49.1 | 74.1 | -357 | 134 | 0.00 |
| O$_2$ (g) | O$_2$ | 1009 | 432 | 659 | 2.62 | 45.5 | 0.00 | 0.00 | 41.9 | 186 | 0.00 | 83.6 | 0.00 | 44.7 | 113 | 215 |
| CO$_2$ (g) | CO$_2$ | -827 | -532 | -650 | 35.2 | 0.00 | 39.7 | 0.00 | -55.5 | -84.8 | 0.00 | -33.1 | 0.00 | 44.6 | -31.7 | -115 |
| No. models found | | 2 | 2 | 2 | 3 | 4 | 2 | 2 | 5 | 2 | 6 | 2 | 1 | 2 | 2 | 1 |

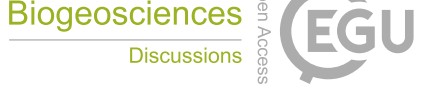

**Table A2.** Pore water processes expressed in mole transfers (µmol l$^{-1}$ day$^{-1}$) as modeled by PHREEQC with pore water data

retrieved at 11 cm below sediment surface. Positive values indicate dissolution, negative values indicate precipitation.

| | | Phase 1. Oxidation (t=0-22) | | | Phase 2. Initial root development (t=22-64) | | | | | | Phase 3. Root influence (t=64-176) | | | | | |
|---|---|---|---|---|---|---|---|---|---|---|---|---|---|---|---|---|
| | | No plant | No plant | No plant | No plant | Plant | No plant | Plant | No plant | Plant | No plant | Plant | No plant | Plant | No plant | Plant |
| **Reactant** | **Composition** | **Mud$_{soft}$** | **Mud$_{sand}$** | **Clay** | **Mud$_{soft}$** | | **Mud$_{sand}$** | | **Clay** | | **Mud$_{soft}$** | | **Mud$_{sand}$** | | **Clay** | |
| **Calcite** | **CaCO$_3$** | 111 | 59.6 | 55.2 | *0.00* | 19.8 | 71.7 | 83.8 | 27.1 | 16.2 | -3.21 | *0.00* | *0.00* | *0.00* | 11.5 | 18.7 |
| **Gypsum** | **CaSO$_4$:2H$_2$O** | -72.5 | -40.7 | -53.4 | *0.00* | -3.81 | *0.00* | *0.00* | *0.00* | *0.00* | *0.00* | *0.00* | *0.00* | *0.00* | *0.00* | *0.00* |
| **Hydroxyapatite** | **Ca$_5$(PO$_4$)$_3$(OH)** | *0.00* | 0.51 | 1.45 | -0.24 | *0.00* | *0.00* | *0.00* | *0.00* | *0.00* | *0.00* | -0.09 | *0.00* | -0.45 | *0.00* | -0.18 |
| **Chalcedony** | **SiO$_2$** | 4.44 | 5.32 | 6.74 | 1.90 | 3.33 | 3.10 | 3.81 | 1.67 | 0.95 | -0.18 | -1.07 | -0.27 | -3.48 | *0.00* | -1.07 |
| **Fe(OH)$_3$ (a)** | **Fe(OH)$_3$** | *0.00* | *0.00* | *0.00* | -0.24 | -6.19 | -41.2 | -51.2 | -25.0 | *0.00* | -0.80 | *0.00* | -1.34 | -23.2 | -13.8 | -8.48 |
| **Pyrite** | **FeS$_2$** | 36.2 | 21.7 | 20.1 | *0.00* | *0.00* | *0.00* | *0.00* | *0.00* | 11.9 | *0.00* | 4.11 | *0.00* | 7.95 | *0.00* | 8.57 |
| **Rhodochrosite** | **MnCO$_3$** | *0.00* | 1.18 | 0.31 | *0.00* | 0.48 | 1.19 | 0.95 | *0.00* | 0.24 | *0.00* | *0.00* | *0.00* | -0.71 | 0.18 | 0.09 |
| **CEC** | **CaX$_2$** | *0.00* | *0.00* | *0.00* | -1.43 | -5.95 | -50.7 | -63.3 | -7.86 | *0.00* | 1.70 | 8.39 | *0.00* | *0.00* | -3.75 | *0.00* |
| | **FeX$_2$** | -35.5 | -20.9 | -19.0 | *0.00* | *0.00* | 42.4 | 51.7 | *0.00* | -11.9 | 1.07 | -3.66 | -0.54 | 15.2 | 4.29 | *0.00* |
| | **KX** | 7.00 | 5.87 | 3.76 | *0.00* | *0.00* | 2.62 | 2.86 | -5.95 | 1.67 | -0.89 | -1.79 | *0.00* | -3.84 | *0.00* | -1.70 |
| | **MgX$_2$** | 15.4 | 13.0 | 4.87 | *0.00* | 4.76 | 7.14 | 8.57 | 8.10 | 7.38 | -1.25 | *0.00* | *0.00* | -4.11 | 2.59 | 5.71 |
| | **NaX** | 25.2 | 9.95 | 24.4 | *0.00* | *0.00* | *0.00* | 5.24 | *0.00* | 6.43 | -4.29 | -4.20 | 1.96 | -12.4 | *0.00* | -5.54 |
| | **NH$_4$X** | 8.12 | *0.00* | *0.00* | 2.86 | 2.62 | *0.00* | -1.67 | 5.48 | 0.95 | 2.23 | -3.39 | *0.00* | -5.80 | -2.77 | -4.20 |
| **H$_2$O (g)** | **H$_2$O x 10$^3$** | *0.00* | *0.00* | *0.00* | *0.00* | *0.00* | *0.00* | *0.00* | *0.00* | *0.00* | *0.00* | *0.00* | *0.00* | -652 | *0.00* | -98.2 |
| **O$_2$ (g)** | **O$_2$** | 119 | 69.5 | 61.9 | *0.00* | *0.00* | *0.00* | *0.00* | *0.00* | 40.5 | *0.00* | 13.6 | *0.00* | 32.6 | *0.00* | 28.4 |
| **CO$_2$ (g)** | **CO$_2$** | 156 | *0.00* | 43.0 | *0.00* | *0.00* | *0.00* | *0.00* | *0.00* | 14.5 | *0.00* | 0.00 | *0.00* | -67.3 | *0.00* | -13.7 |
| | **No. models found** | 2 | 2 | 1 | 4 | 4 | 2 | 2 | 3 | 2 | 1 | 4 | 2 | 4 | 2 | 1 |

629