# Peer review of "Wetland eco-engineering: measuring and modeling feedbacks of oxidation"

_Biogeosciences, 2016_

## Referee Comment (RC1) · Anonymous Referee #1 · 31 May 2016

General Comments: Understanding the impacts of vegetation on the biogeochemical cycling of nutrients is important, especially in newly created systems, and this manuscript helps to examine how these cycles differ based on different substrates. When discussing the systems, the use of the term "building material" to refer to sediment seems a bit odd. Maybe something along the lines of "introduced substrate/sediment" or "donor substrate/sediment" would be more appropriate. The introduction, while it includes aims, is lacking clearly defined, testable hypotheses. Overall, the entire manuscript needs more information on statistical analyses, particularly what tests were used to find the p-values listed (mainly in the tables), as well as additional p-values throughout the Results and Discussion section.

Specific Comments: Need a statistical analysis section within the Methods to outline the statistical program that was used, the statistical analyses that were preformed and any data transformations that were necessary. Lines 191-195 (1st paragraph of Results and Discussion): Paragraph would be better suited for the Methods section of the paper. Need to include more p-values within the text, even if they are not significant, when you mention a treatment being different from another. When writing p-values make sure you include some information on the test that you used for the analysis. The size of the pots used to grow Phragmites australis seems small for the size of the plant. Do you have any information on whether or not the plants had become pot bound, which could cause wilting, discoloration of leaves, and stunted growth? The text in the figures and some of the tables is difficult to read due to font size. Figures 1, 2, 3, 4, & 5: If possible, make the 3 soil treatment labels (above graph columns or in legends) larger. Figures 3, 4, & 5: Mudsand and Clay bars are difficult to distinguish in black and white, it may be better to choose a pattern or solid color that would provide more contrast.

Technical Corrections: Line by line technical corrections include: Line 95: remove "plants of" and make sure wording is either all singular or all plural; Line 120: Typo, "form" in this sentence should be "from"; Line 418: Make sure you are consistent in using either all singular or all plural words; Lines 425-426 "in the future wetland" change to "in created wetlands".

---

## Referee Comment (RC2) · I. Bauer (Referee) · 11 Jul 2016

This manuscript examines the biogeochemical feedback between vegetation and soil, specifically in soft, dredged sediments. Ultimately, this paper and similar work will enable designers of these created wetland habitats to select plants that will aid in accelerating ecosystem development created using such soils. With the increasing desire world-wide to restore wetlands for their many natural benefits, and the potential to aid existing wetlands in the race against sea level rise, these results and results of studies using this methodology can have great benefit to the ecosystem restoration community.

Overall, the study is informative and well written, though a few language improvements and additional explanation on a couple points would enhance the reader's background and general understanding. A few questions/comments on specific points in the manuscript– *** Sentence spanning lines 64-66: Agreed that the roots may enhance consolidation processes by increasing drainage. Did you also consider increased consolidation through evapotranspiration? *** Line 115: Why was Dorsilit selected? Recommend providing a few additional details about its properties. *** Paragraph beginning on line 180: Were these timeframes identified through research and then used? Or were they identified during this study? Please clarify this point. If identified through the former method, make sure to cite references; if the latter, provide a few details on how the stages were differentiated. Recommend changing the word "used" in line 180 to "identified" which would be accurate whether it was identified through literature or during the study. *** Paragraph beginning on line 280: Do you feel the aeration occurring at D11 would also occur in situ, when the soil extends further from the plant roots, or is it possible that this occurred due to the close boundary with the container? Were any decisions made about the set-up of the experiment to reduce such boundary influences? *** Paragraph beginning line 286: You use the phrase "some differences" were noted, but then only mention one specific difference. Consider summarizing other differences you wish to highlight or referring to the differences discussed earlier in the section. *** Sentence spanning lines 399-401: For the additional studies/testing you recommend, would you recommend this be done in-situ or using the methodology developed during this study? Recommend including a few additional details to this point. *** Sentence spanning 420-422: Recommend emphasizing whether the impact was positive or negative. Also, do you feel the results show wetland creators should add sand or not? *** In Table 2, consider listing clay first so that its composition can easily be compared to the soft mud. *** In Figure 4, it appears the results for soft mud are significantly different from the results for clay–should soft mud still have both b and a indicators?

Additional language and typographical recommendations: *** Lines 19-20: Recommended wording of last half of sentence–". . .is an example; here, dredging some of the... will soon begin." (More direct wording.) *** Lines 26-27: The subject of the

first part of the sentences is N:P ratios, and I believe this is not the subject of the portion after "and." Insert appropriate subject between "and" and "were affected," potential suggestions include plants, plant health, plant growth, etc. *** Line 27: Insert a comma– "...uptake of N, but by..." *** Line 35: Use "be used" instead of "are used," or restructure sentence to read "Given these two feedback mechanisms, we propose the use of Fe-tolerant species rather than species that thrive in N-limited conditions." *** Line 45: Insert a comma after "Nowadays." *** Throughout, but noted on line 54: I was a little uncertain whether "soft clay-rich" was referring to a soil rich in soft clay or one that was rich in clay and also soft. If the former, consider using "soft-clay-rich," if the latter, change to "soft, clay-rich." *** Lines 54-55: Restructure sentence: "In the Netherlands, a soft-clay-rich lake-bed sediment is causing serious turbidity problems in the Markermeer (and artificial like of 691 km^2)." *** Line 58: Recommend "plans are underway" instead of "it is planned." *** Line 69: insert comma after "formation." *** Line 69: Believe should use "signs" instead of "sign." *** Lines 73-74: Recommend the following after the comma: "it is essential to determine which eco-engineer is most appropriate for accelerating ecosystem development in these protosoils." *** Line 74: I am unfamiliar with "protosoils," but that may just be my background, consider whether this is a common term for others in the industry and change or explain if appropriate. *** Lines 79-80: Recommend rewording last sentence as follows: "Two types of clay-rich deposits are the indented building material for the wetlands." *** Line 80: Recommend changing beginning of the sentence use "their presence is" or "their composition is" in place of "they are." *** Line 80: "Products" should be singular because it refers to "a combination," which is also singular. *** Sentence lines 91-94: Recommend moving "We set up... pore water," to the beginning of the sentence for added clarity. *** Line 153: add "content" after "Nitrogen." *** Line 191: Delete "below" and begin the sentence with "First." Also add a comma after "First." *** Line 202: use "than" instead of "then." *** Line 227: insert a comma between "without plants" and "the." *** Line 234: Believe the reference should be to Figure 1g rather than Figure 2g. *** Line 242: Delete "it must be taken into account that" (More direct wording.) *** Lines 268 and 271: Consider indenting the chemical equations. *** Line 272: Appears to be an unintended blank line after the equation (2). If line 273 is a new paragraph, indent it; if it is a continuation of previous paragraph, simply delete blank line. *** Line 284-285: Recommend moving this sentence up to be a part of the previous paragraph. *** Lines 292-293: Recommend providing clarity by rewording to say "While the pore water compositions did not show clear differences between unplanted and planted conditions during the initial stage of plant growth, . . ." *** Line 347: Add a comma after "the experiment." *** Line 411: Add comma after "plant growth." *** Line 412: Change to ". . .promotes P mobilization, enhancing plant growth." *** Line 416: Delete comma after "P uptake." *** Line 422: Consider whether using "of" rather than "on" would be more appropriate. *** Line 423: Insert comma after "detail." *** I believe it is customary to eliminate the use of "we" and "our" in scientific papers, though I know it is difficult to do. *** Line 608: delete comma after "(a-c)."

---

## Author Comment (AC1) · 22 Jul 2016

**Reply to Anonymous referee #1**

We would like to thank anonymous reviewer #1 for giving useful feedback on our manuscript. We have responded in red.

**General Comments:**

Understanding the impacts of vegetation on the biogeochemical cycling of nutrients is important, especially in newly created systems, and this manuscript helps to examine how these cycles differ based on different substrates.

We thank the reviewer for this comment and for acknowledging the importance of our study.

When discussing the systems, the use of the term "building material" to refer to sediment seems a bit odd. Maybe something along the lines of "introduced substrate/sediment" or "donor substrate/sediment" would be more appropriate.

Although the term "building material" is used within the Building with Nature community, we agree with the referee that for a better understanding – and to comply with a more international standard – another term should be used. In our manuscript, we changed "building material" into "situated sediment". This is changed throughout the manuscript.

The introduction, while it includes aims, is lacking clearly defined, testable hypotheses.

We added a paragraph in the introduction defining our hypothesis:

*"Changes in biogeochemical processes that are related to oxidation are expected to play a major role as* P. australis *is known for its high radial oxygen loss (Brix et al., 1996; Dickopp et al., 2011; Smith and Luna, 2013). Oxidation of the sediment will decrease the concentration of phytotoxins typically found in waterlogged soils, such as iron, and therefore will have a positive effect on plant development. This will be more pronounced in undisturbed mud, which is largely anoxic, than disturbed mud, of which the top layer is already oxidized and where bioturbation modified the sediment. The type of biogeochemical processes altered will depend on the intrinsic properties of the different sediment types, which will be examined in this study."*

Overall, the entire manuscript needs more information on statistical analyses, particularly what tests were used to find the p-values listed (mainly in the tables), as well as additional p-values throughout the Results and Discussion section. Need a statistical analysis section within the Methods to outline the statistical program that was used, the statistical analyses that were preformed and any data transformations that were necessary. Need to include more p-values within the text, even if they are not significant, when you mention a treatment being different from another. When writing p-values make sure you include some information on the test that you used for the analysis.

We added an extra section within the Methods (2.4) explaining the programs and tests we used to detect significant differences between sediment treatments:

*"Statistical analysis was carried out using the programs R and SPSS. Differences in sediment, pore water and plant tissue concentrations between sediment treatments were determined using one-way ANOVA with a Tukey's honestly significant difference (HSD) post hoc test. No statistics could be applied to the mineralogical sediment composition (XRD analysis) due to absence of replicates."*

Throughout the ms we added p-values, especially in section 3.1 and 3.4.

Lines 191-195 (1st paragraph of Results and Discussion): Paragraph would be better suited for the Methods section of the paper.

This paragraph presents how the Results and Discussion sections is ordered and enhances the readability of the manuscript (acts as a reading guide). We prefer leaving that paragraph at the beginning of the Results and Discussion section.

The size of the pots used to grow Phragmites australis seems small for the size of the plant. Do you have any information on whether or not the plants had become pot bound, which could cause wilting, discoloration of leaves, and stunted growth?

We are aware that a root-bound effect can lead to stunted growth and damage to foliage (see Ray and Sinclair (1998) and Townend and Dickinson (1995) for a description of these effects). In our case, we are sure that *P. australis* did not became root bound: the roots did not stick to the walls, nor were they clotted (see photo of roots directly after harvest). The root biomass was in all cases lower than 4 gr dw with a pot volume of c. 1400 $cm^3$. Moreover, Townend and Dickinson (1995) reported that root-bound effects of plants in pots with the same size do not occur in the first 150 days after transplantation, which is about the same as the duration of our experiment (176 days).

[Figure]

The text in the figures and some of the tables is difficult to read due to font size. Figures 1, 2, 3, 4, & 5: If possible, make the 3 soil treatment labels (above graph columns or in legends) larger. Figures 3, 4, & 5: Mudsand and Clay bars are difficult to distinguish in black and white, it may be better to choose a pattern or solid color that would provide more contrast.

The design of Figures 1, 2, 3, 4 and 5 are now changed. We took into account the recommendation made by referee #1.

**Technical Corrections:**

Line 95: remove "plants of" and make sure wording is either all singular or all plural;

Line 120: Typo, "form" in this sentence should be "from";

Line 418: Make sure you are consistent in using either all singular or all plural words;

Lines 425-426: "in the future wetland" change to "in created wetlands".

Thank you. The technical corrections above are all implemented in the manuscript.

**References used in this reply**

Ray JD, Sinclair TR (1998). The effect of pot size on growth and transpiration of maize and soybean during water deficit stress. Journal of Experimental Botany 325: 1381-1386.

Townend J, Dickinson AL (1995). A comparison of rooting environments in containers of different sizes. Plant and Soil 175: 139-146.

---

## Author Response (AR1)

**Authors response**

We would like to thank anonymous referee 1 and Ingrid Bauer for their helpful comments on the manuscript. Below, you will find our point-to-point reply of all the comments raised as well as the improved manuscript.

**ANONYMOUS REFEREE 1**
**General Comments:**

When discussing the systems, the use of the term "building material" to refer to sediment seems a bit odd. Maybe something along the lines of "introduced substrate/sediment" or "donor substrate/sediment" would be more appropriate.

Although the term "building material" is used within the Building with Nature community, we agree with the referee that for a better understanding – and to comply with a more international standard – another term should be used. In our manuscript, we changed "building material" into "situated sediment" (lines: 39 and 462) or we changed the sentence to explain better (lines 17, 86-87, 134 and 489).

The introduction, while it includes aims, is lacking clearly defined, testable hypotheses.

In the revised manuscript we added a paragraph to the introduction defining our hypothesis

**Lines 111-119:**

*"Changes in biogeochemical processes that are related to oxidation are expected to play a major role as* P. australis *is known for its high radial oxygen loss (Brix et al., 1996; Dickopp et al., 2011; Smith and Luna, 2013). Oxidation of the sediment will decrease the concentration of phytotoxins typically found in waterlogged soils, such as iron, and therefore will have a positive effect on plant development. This will be more pronounced in undisturbed mud, which is largely anoxic, than in disturbed mud, of which the top layer is already oxidized and where bioturbation modified the sediment. The type of biogeochemical processes altered will depend on the intrinsic properties of the different sediment types, which will be examined in this study."*

Overall, the entire manuscript needs more information on statistical analyses, particularly what tests were used to find the p-values listed (mainly in the tables), as well as additional p-values throughout the Results and Discussion section. Need a statistical analysis section within the Methods to outline the statistical program that was used, the statistical analyses that were preformed and any data transformations that were necessary. Need to include more p-values within the text, even if they are not significant, when you mention a treatment being different from another. When writing p-values make sure you include some information on the test that you used for the analysis.

We added an extra section within the Methods (2.4) explaining the programs and tests we used to detect significant differences between sediment treatments.

**Lines 221-226:**

*"Statistical analysis was carried out using the programs R and SPSS. Differences in sediment, pore water and plant tissue concentrations between sediment treatments were determined using one-way*

*ANOVA with a Tukey's honestly significant difference (HSD) post hoc test. No statistics could be applied to the mineralogical sediment composition (XRD analysis) due to absence of replicates."*

**Lines 242, 247, 257 and 400**
We added p-values.

Lines 191-195 (1st paragraph of Results and Discussion): Paragraph would be better suited for the Methods section of the paper.

This paragraph presents how the Results and Discussion section is ordered and enhances the readability of the manuscript (acts as a reading guide). We prefer leaving that paragraph at the beginning of the Results and Discussion section.

The size of the pots used to grow Phragmites australis seems small for the size of the plant. Do you have any information on whether or not the plants had become pot bound, which could cause wilting, discoloration of leaves, and stunted growth?

We are aware that a root-bound effect can lead to stunted growth and damage to foliage (see Ray and Sinclair (1998) and Townend and Dickinson (1995) for a description of these effects). In our case, we are sure that *P. australis* did not became root bound: the roots did not stick to the walls, nor were they clotted (see photo of roots directly after harvest). The root biomass was in all cases lower than 4 gr dw with a pot volume of c. 1400 $cm^3$. Moreover, Townend and Dickinson (1995) reported that root-bound effects of plants in pots with the same size do not occur in the first 150 days after transplantation, which is about the same as the duration of our experiment (176 days).

[Figure]

The text in the figures and some of the tables is difficult to read due to font size. Figures 1, 2, 3, 4, & 5: If possible, make the 3 soil treatment labels (above graph columns or in legends) larger. Figures 3, 4, & 5: Mudsand and Clay bars are difficult to distinguish in black and white, it may be better to choose a pattern or solid color that would provide more contrast.

To overcome these shortcomings, the designs of Figures 1, 2, 3, 4 and 5 are now changed:

[Figure]

[Figure]

Figure 2

[Figure]

Figure 3

[Figure]

Figure 4

[Figure]

Figure 5

**Technical Corrections:**

Line 95: remove "plants of" and make sure wording is either all singular or all plural;

Line 120: Typo, "form" in this sentence should be "from";

Line 418: Make sure you are consistent in using either all singular or all plural words;

Lines 425-426: "in the future wetland" change to "in created wetlands".

Thank you. The technical corrections above are all implemented in the manuscript.

**REFEREE 2: INGRID BAUER**

\*\*\* Sentence spanning lines 64-66: Agreed that the roots may enhance consolidation processes by increasing drainage. Did you also consider increased consolidation through evapotranspiration?

In principle, the hydraulic design of a plant root follows that of a porous pipe (Zwieniecki et al., 2003). Hence, plant roots potentially drain soils by extracting water, of which 97% is lost through the leaves (i.e. plant transpiration) (Sinha, 2004). Therefore, we do not differentiate between increasing drainage and (evapo)transpiration, which are treated by us as one process.

\*\*\* Line 115: Why was Dorsilit selected? Recommend providing a few additional details about its properties.

We mixed the soft mud with sand to see how increasing the grain size would affect biogeochemical processes in the soil. This is relevant because there are numerous cases were mud was mixed with sand to enhance soil stability. However, since we are interested in the influence of grain size on biogeochemical processes, it is important that the chemical properties of the sand itself do not influence these biogeochemical processes. Therefore, we selected Dorsilit, which is sand that consists almost exclusively of unreactive crystal silica (c. 99% $SiO2$), with the remaining part consisting of aluminum oxide (c. 0.6% $Al_2O_3$). The grains are 0.3-0.8 mm in size with a median diameter (D50) of 0.57 mm. We added information about the grain size of this material at **line 136-137**:

*"…a 1:1 mixture was made by mixing mud with Dorsilit® crystal silica sand (c. 99% SiO2) which had been autoclaved for one hour at 120 °C prior to mixing. The sand grains of this material are 0.3-0.8 mm in diameter with D50 being 0.57 mm."*

\*\*\* Paragraph beginning on line 180: Were these timeframes identified through research and then used? Or were they identified during this study? Please clarify this point. If identified through the former method, make sure to cite references; if the latter, provide a few details on how the stages were differentiated. Recommend changing the word "used" in line 180 to "identified" which would be accurate whether it was identified through literature or during the study.

The first time frame represents the period where no plants were growing in the pot – i.e. before transplantation. Time frame 2 and 3 were *identified* during this study by looking at the measured data at 1 and 11 cm depth. After transplantation, it takes a while before the roots have sufficient biomass to start interfering with the biogeochemical processes in the deeper part of the soil. The time it takes before sufficient biomass is produced is highly species specific, so no attempt was made to identify this period through literature research. When pore water chemistry at 11 cm depth in the planted condition started to deviate from the unplanted condition, we regarded that as a sign that roots were influencing the biogeochemical processes in this part of the soil. For some chemical variables this was the case after 64 days, for other chemical variables it was after 92 days (see Figures 1 and 2). Therefore, we chose the more conservative period of t=64-176 as the time frame where roots influenced these processes. Furthermore, *P. australis* is known for its high radial oxygen loss, so oxidation processes at 11 cm depth are largely expected in the third time frame for the planted condition. This is also what the model calculated (Table 3). We added a few sentences in this paragraph for clarification **at lines 211-214**:

*"These time frames were identified by analysing the chemical data that was collected. When concentrations at D11 in the planted condition started to deviate from the unplanted condition, this was seen as a sign that plant roots started to influence pore water chemistry."*

*** Paragraph beginning on line 280: Do you feel the aeration occurring at D11 would also occur in situ, when the soil extends further from the plant roots, or is it possible that this occurred due to the close boundary with the container? Were any decisions made about the set-up of the experiment to reduce such boundary influences?

Numerous studies have shown that radial oxygen loss by *P. australis* in anoxic soils oxidize the rhizosphere (e.g. Armstrong and Armstrong, 2001; Armstrong et al., 2006; Tercero et al., 2015). In our experimental design, we decided to maintain a water level of 9 cm, which is 3 cm *above* D11. D11 was at all times submerged which prevented oxygen to penetrate: De Lucas Pardo (2014) showed that oxygen could only penetrate the first 2 mm's in soft mud and clay from lake Markermeer in submerged conditions illustrating the reduction capacity of the materials studied. Furthermore, the rhizons extracted pore water 5 cm from the pot wall (at the center of the pot). The water level and the placement of the rhizons are described in **lines 141-148** but we added extra information now, to clarify this better for the readers. The paragraph now reads as follows:

*"Plastic pots (diameter 10 cm, depth 18 cm) with a perforated base were filled to within 1 cm from the top with one of the three sediment types used (t = 0). In each pot, two soil moisture samplers (Rhizon Flex-5cm; Rhizosphere, Wageningen, the Netherlands) were installed horizontally at depths of 1 cm and 11 cm below the sediment surface (these depths are hereafter referred to as $D_1$ and $D_{11}$), its tip reaching 5 cm from the pot wall. The pots were stood in rows in the basin. The water level was maintained at 9 cm so that the sediment at $D_{11}$ remained saturated while the sediment at $D_1$ could oxidize and dry. Each sediment type had 13 replicates."*

*** Paragraph beginning line 286: You use the phrase "some differences" were noted, but then only mention one specific difference. Consider summarizing other differences you wish to highlight or referring to the differences discussed earlier in the section.

Since we want to highlight the difference explained in these lines, we changed that sentence to avoid confusion.

*Lines 337-340:*
*"The processes described above occurred in all three sediments, although oxidation was higher in $Mud_{soft}$ than in $Mud_{sand}$ and Clay, probably because higher evaporation rates in $Mud_{soft}$ enhanced oxidation and affected other reactants related to oxidation."*

\*\*\* Sentence spanning lines 399-401: For the additional studies/testing you recommend, would you recommend this be done in-situ or using the methodology developed during this study? Recommend including a few additional details to this point.

Thank you, we added some points to this paragraph. To come up with a sound hypothesis/prediction with respect to ecosystem development/feedback mechanisms on the constructed wetlands, an in-situ experiment should be carried out as a number of other factors are in play that are not tested ex-situ (e.g. wave action, wind). However, such an experiment can only be carried out when the crest has stabilized sufficiently on the constructed wetland. Ex-situ testing enables us to focus more on specific interactions. We made this point more clear by adding the following (**lines 467-470**):

"*Not all environmental factors that potentially interfere with the processes and feedbacks described in this study could be taken into account with this experimental design (e.g. wave action, wind). Therefore, we recommend to carry out experiments on the wetlands themselves once the crest has stabilized sufficiently.*"

\*\*\* Sentence spanning 420-422: Recommend emphasizing whether the impact was positive or negative. Also, do you feel the results show wetland creators should add sand or not?

We changed line 420-422 as follows (now spanning **lines 490-493**):

"*However, when the mud is mixed with sand, the enhanced aeration due to the change in grain-size composition results in higher oxidation rates, increasing the impact of the positive feedback mechanisms involving P mobilization and iron toxicity.*"

From a physical perspective, it is clear that mud mixed with sand would enhance consolidation and crest stability. However, in practical mixing sand with mud is more expensive. Given the difference in plant development between $Mud_{soft}$ and $Mud_{sand}$ we do not recommend adding sand to the mud in the amount we did in the experiment (Figure 3).

*** In Table 2, consider listing clay first so that its composition can easily be compared to the soft mud.

Thank you for this valuable suggestion. We changed Table 2 as follows:

Table 2

| | Unit | n per type | Clay Mean | SD | Mud$_{soft}$ Mean | SD | Mud$_{sand}$ Mean | SD |
|---|---|---|---|---|---|---|---|---|
| *Aqua regia / CS* | | | | | | | | |
| **Al*** | mg/kg | 15 | 21989 | *4512* | 16593 | *3130* | 6394 | *2439* |
| **Ca** | mg/kg | 15 | 48031 | *3032* | 45635 | *6020* | 18877 | *3572* |
| **Fe*** | mg/kg | 15 | 27766 | *3764* | 20745 | *2987* | 7804 | *2281* |
| **K** | mg/kg | 15 | 5371 | *1262* | 4102 | *641* | 1723 | *742* |
| **Mg*** | mg/kg | 15 | 8041 | *1017* | 6636 | *906* | 2531 | *558* |
| **Mn*** | mg/kg | 15 | 710 | *166* | 577 | *160* | 238 | *62* |
| **Na*** | mg/kg | 15 | 992 | *379* | 526 | *158* | 219 | *64* |
| **P*** | mg/kg | 15 | 1186 | *217* | 649 | *169* | 259 | *56* |
| **S** | mg/kg | 15 | 5727 | *710* | 5586 | *698* | 3001 | *846* |
| **Sr** | mg/kg | 15 | 148 | *21* | 135 | *26* | 62 | *14* |
| **Ti** | mg/kg | 15 | 312 | *74* | 312 | *77* | 125 | *44* |
| **Zn*** | mg/kg | 15 | 159 | *58* | 110 | *29* | 43 | *18* |
| | | | | | | | | |
| *Seq. P extraction* | | | | | | | | |
| **Exchangeable P** | mg/kg | 15 | 14.3 | *6.81* | 11.9 | *3.50* | 5.9 | *1.79* |
| **Fe- bound P*** | mg/kg | 15 | 772 | *263* | 279 | *61.7* | 94.5 | *29.0* |
| **Ca-bound P** | mg/kg | 15 | 146 | *43.3* | 121 | *30.9* | 36.8 | *13.1* |
| **Detrital P** | mg/kg | 15 | 147 | *16.5* | 169 | *14.1* | 51.5 | *10.9* |
| **Organic P** | mg/kg | 15 | 99.6 | *20.0* | 117 | *25.1* | 47.7 | *8.38* |
| | | | | | | | | |
| *XRD* | | | | | | | | |
| **Quartz** | % | 1 | 48 | | 37 | | n.a. | |
| **Calcite** | % | 1 | 9 | | 9 | | n.a. | |
| **Pyrite** | % | 1 | 0.6 | | 0.6 | | n.a. | |
| **Illite** | % | 1 | 15 | | 21 | | n.a. | |
| **Smectite** | % | 1 | 11 | | 14 | | n.a. | |
| **Kaolinite** | % | 1 | 3 | | 5 | | n.a. | |
| **Chlorite** | % | 1 | 2 | | 3 | | n.a. | |
| | | | | | | | | |
| *Other* | | | | | | | | |
| **Organic matter** | % | 5 | 6.7 | *0.6* | 7.2 | *0.6* | 2.8 | *0.4* |
| **CEC (calculated)** | meq/100g | | 30.0 | | 37.2 | | 12.4 | |

*** In Figure 4, it appears the results for soft mud are significantly different from the results for clay–should soft mud still have both b and a indicators?

No mistake was made in Figure 4. The difference between clay and soft mud for N is not significant ($p = 0.051$).

Additional language and typographical recommendations:

All the recommendations below are implemented in the revised version of the manuscript with one exception (outlined in red).

\*\*\* Lines 19-20: Recommended wording of last half of sentence–". . .is an example; here, dredging some of the... will soon begin." (More direct wording.)

\*\*\* Lines 26-27: The subject of the first part of the sentences is N:P ratios, and I believe this is not the subject of the portion after "and." Insert appropriate subject between "and" and "were affected," potential suggestions include plants, plant health, plant growth, etc.

\*\*\* Line 27: Insert a comma– "...uptake of N, but by..."

\*\*\* Line 35: Use "be used" instead of "are used," or restructure sentence to read "Given these two feedback mechanisms, we propose the use of Fe-tolerant species rather than species that thrive in N-limited conditions."

\*\*\* Line 45: Insert a comma after "Nowadays."

\*\*\* Throughout, but noted on line 54: I was a little uncertain whether "soft clay-rich" was referring to a soil rich in soft clay or one that was rich in clay and also soft. If the former, consider using "soft-clay-rich," if the latter, change to "soft, clay-rich."

\*\*\* Lines 54-55: Restructure sentence: "In the Netherlands, a soft-clay-rich lake-bed sediment is causing serious turbidity problems in the Markermeer (and artificial like of 691 km^2)."

\*\*\* Line 58: Recommend "plans are underway" instead of "it is planned."

\*\*\* Line 69: insert comma after "formation."

\*\*\* Line 69: Believe should use "signs" instead of "sign."

\*\*\* Lines 73-74: Recommend the following after the comma: "it is essential to determine which eco-engineer is most appropriate for accelerating ecosystem development in these protosoils."

\*\*\* Line 74: I am unfamiliar with "protosoils," but that may just be my background, consider whether this is a common term for others in the industry and change or explain if appropriate.

\*\*\* Lines 79-80: Recommend rewording last sentence as follows: "Two types of clay-rich deposits are the indented building material for the wetlands."

\*\*\* Line 80: Recommend changing beginning of the sentence use "their presence is" or "their composition is" in place of "they are."

\*\*\* Line 80: "Products" should be singular because it refers to "a combination," which is also singular.

\*\*\* Sentence lines 91-94: Recommend moving "We set up. . . pore water," to the beginning of the sentence for added clarity.

\*\*\* Line 153: add "content" after "Nitrogen."

\*\*\* Line 191: Delete "below" and begin the sentence with "First." Also add a comma after "First."

\*\*\* Line 202: use "than" instead of "then."

\*\*\* Line 227: insert a comma between "without plants" and "the."

\*\*\* Line 234: Believe the reference should be to Figure 1g rather than Figure 2g.

\*\*\* Line 242: Delete "it must be taken into account that" (More direct wording.)

\*\*\* Lines 268 and 271: ConC3 BGD Interactive comment Printer-friendly version Discussion paper sider indenting the chemical equations.

\*\*\* Line 272: Appears to be an unintended blank line after the equation (2). If line 273 is a new paragraph, indent it; if it is a continuation of previous paragraph, simply delete blank line.

*** Line 284-285: Recommend moving this sentence up to be a part of the previous paragraph.

*** Lines 292-293: Recommend providing clarity by rewording to say "While the pore water compositions did not show clear differences between unplanted and planted conditions during the initial stage of plant growth, . . ."

*** Line 347: Add a comma after "the experiment."

*** Line 411: Add comma after "plant growth."

*** Line 412: Change to ". . .promotes P mobilization, enhancing plant growth."

*** Line 416: Delete comma after "P uptake."

*** Line 422: Consider whether using "of" rather than "on" would be more appropriate.

*** Line 423: Insert comma after "detail."

*** I believe it is customary to eliminate the use of "we" and "our" in scientific papers, though I know it is difficult to do.

Answer: We believe there is no general rule anymore of (de-)personalization of scientific papers. It seems to be a matter of taste. We chose not to change our wording, since we think it is concise and direct as it is now.

*** Line 608: delete comma after "(a-c)."

**References used in this reply**

Armstrong J, W Armstrong (2001). Rice and *Phragmites*: effects of organic acids on growth, root permeability, and radial oxygen loss to the rhizosphere. American Journal of Botany 88: 1359-1370.

Armstrong J, RE Jones, W Armstrong (2006). Rhizome phyllosphere oxygenation in Phragmites and other species in relation to redox potential, convective gas flow, submergence and aeration pathways. New Phytologist 172: 719–731.

De Lucas Pardo MA (2014). Effect of biota on fine sediment transport processes. A study of lake Markermeer. PhD dissertation, Delft University.

Ray JD, Sinclair TR (1998). The effect of pot size on growth and transpiration of maize and soybean during water deficit stress. Journal of Experimental Botany 325: 1381-1386.

Sinha RK (2004). Modern plant physiology. Chapter 5: loss of water from plants (transpiration, guttation, exudation). Alpha Science Internation Ltd., Pangbourne, United Kingdom.

Tercero MC, Álvarez-Rogel J, Conesa HM, Ferrer MA, Calderón AA, López-Orenes A, González-Alcaraz MN (2015). Response of biogeochemical processes of the water-soil-plant system to experimental flooding-drying conditions in a eutrophic wetland: the role of Phragmites australis. Plant and Soil 396:109-125.

Townend J, Dickinson AL (1995). A comparison of rooting environments in containers of different sizes. Plant and Soil 175: 139-146.

Zwieniecki MA, Thompson MV, Holbrook NM (2003). Understanding the hydraulics of porous pipes: tradeoffs between water uptake and root length utilization. Journal of Plant Growth Regulation 21: 315-323.

[revised manuscript text omitted]

---

## Author Response (AR2)

**Authors response**

We would like to thank the anonymous referee for his/her helpful comments on the manuscript. Below, you will find the issues raised by the referee with our response (in red).

**ANONYMOUS REFEREE**

General Comments:
When including p-values in the results, write out the p-value for each group/type examined, avoid using ranges.

We agree with the referee that the results can be clarified by writing out p-values. Ranges were used in lines 247 and 257. Because these ranges refer to the same p-values, we removed the range in line 247 and wrote out the p-value for each group in line 257.

Figure 4: The red bar on the graph is unaccounted for in the legend/figure caption.
Figure 5: The red bar on the graph is unaccounted for in the legend/figure caption.

Thank you for this comment. Despite our care revising these figures, we missed this error. We corrected the legends in both Figure 4 and 5. The red color added to the legends represents Clay.

Technical Corrections:
Line 65: Change "wetland" to "wetlands"
Line 104: Change "eco-engineer" to "eco-engineers"

Thank you. The technical corrections were taken care off as suggested.